# Current Physical Therapy for Skin Scar Management: A Scoping Review

**DOI:** 10.3390/jcm14175920

**Published:** 2025-08-22

**Authors:** Sara Di Serio, Matteo Congiu, Silvia Minnucci, Valentina Scalise, Firas Mourad

**Affiliations:** 1Department of Clinical Science and Translational Medicine, University of Rome Tor Vergata, 00133 Rome, Italysilviaminnucci8@gmail.com (S.M.); scalise.valentina@gmail.com (V.S.); 2Department of Neuromotor Rehabilitation, IRCCS Istituto Auxologico Italiano, 20145 Milan, Italy; 3Health Department, LUNEX University of Applied Sciences, 4671 Differdange, Luxembourg; 4Luxembourg Health & Sport Sciences Research Institute A.s.b.l., 50, Avenue du Parc des Sports, 4671 Differdange, Luxembourg; 5Facoltà Dipartimentale di Medicina e Chirurgia, Università Campus Bio-Medico di Roma, 00128 Rome, Italy

**Keywords:** cicatrix, keloid, burns, rehabilitation, conservative treatment, musculoskeletal manipulations

## Abstract

**Background:** Scar impairments impose a significant economic burden and negatively impact an individual’s well-being and quality of life. However, there is a lack of standardization in physical therapy interventions for scar management. **Objective:** This study aimed to provide a comprehensive overview of studies addressing non-invasive physical therapy interventions for scar management. **Methods:** This scoping review was conducted following the Joanna Briggs Institute (JBI) Manual for Evidence Synthesis. Six databases were searched, and additional studies were retrieved through gray literature and the reference lists of included articles. All studies considering non-invasive physical therapy interventions for scar management were included. No restrictions were applied regarding time, context or publication type. Results were illustrated using descriptive statistics and summarized in an infographic. **Results:** Out of 13,419 initial records, 92 studies met the inclusion criteria. Most articles were narrative reviews (*n* = 41) followed by randomized controlled trials (RCT) (*n* = 18). The most reported interventions were pressure therapy (*n* = 41), physical therapy modalities (*n* = 37), silicone-based products (*n* = 29) and massage (*n* = 20). **Conclusions:** Scar management involves a wide range of physical interventions. However, research has predominantly focused on adults, particularly those with burns, with limited attention given to pediatric or non-adult populations. Furthermore, there is significant variability in the application parameters, scar localization and size. Examining the included study designs, most of the research presented reduced sample sizes and lacked control groups. Notably, almost half of the studies were based on expert opinions. Future high-quality research is needed to identify evidence-based interventions for the clinical management of scars.

## 1. Introduction

Scar-related impairments affect approximately 100 million individuals every year in high-income countries [1]; of these, around 11 million will develop keloids [2]. In low- and middle-income countries, scars also represent a substantial burden, with limited access to specialized care and rehabilitation services contributing to worse functional outcomes, greater disability, and socioeconomic impact [3]. Scar occurrence and irritation are frequently attributed to trauma, burns, surgery, vaccination, skin piercing, acne, and herpes zoster. Additionally, these conditions are often accompanied by intermittent pain, persistent itching, and a feeling of constriction [4]. Under normal circumstances, an immature scar undergoes a maturation process that may last several months. However, the maturation process may be compromised as the inflammatory process continues within the scar. As a result, the immature stage is prolonged, potentially resulting in pathological scars such as keloids and hypertrophic scars (HS) [5]. Mechanical forces, genetics, lifestyle choices, hormonal factors like estrogen, and conditions such as hypertension and pregnancy may contribute to the development of pathological scars [5]. Pathological scars are mainly divided into keloids and HS and have been observed to be associated with negative physical and psychological consequences such as scar contracture, a limited range of motion, increased pain, itching, anxiety, and a decreased quality of life [6,7]. Although growing evidence highlights their impact on quality of life, the burden of scars is an inadequately addressed topic, and most available studies focus on pharmacological or surgical approaches. Recent studies on scar management have highlighted the positive role that physical interventions may play in scar care. However, an overview of the various interventions is still needed. Two previous systematic reviews [8,9] found the positive effects of massage, lotions, silicone, splinting/casting, and modalities (e.g., extracorporeal shockwave therapy), but neither included common interventions such as patient education, manual therapy, or exercise. Thus, the heterogeneity among protocols adopted in clinical trials limits the generalizability of results.

Given that the primary intended audience of this review includes clinicians such as physiotherapists involved in scar management, we emphasized dosage, application parameters, and clinical implications to support evidence-based decision-making in daily practice, while still providing sufficient methodological detail for research purposes. Therefore, the present scoping review aims to identify and map the available evidence and analyze knowledge gaps on this topic as a precursor for future research [10]. In particular, we aimed to provide a comprehensive overview of all studies addressing physiotherapy and non-invasive interventions for skin scar management.

## 2. Materials and Methods

Our scoping review was conducted in accordance with the JBI Manual for Evidence Synthesis [11]. The six-stage methodology suggested by Arksey and O’Malley was followed [12]. The Preferred Reporting Items for Systematic Reviews and Meta-Analyses Extension for Scoping Reviews (PRISMA-ScR) Checklist was used for reporting [13]. The scoping review protocol was registered in MedRxiv [14] and can be retrieved at the following URL: https://doi.org/10.1101/2024.05.14.24307367.


*Review Questions*



*What is known from the existing literature about physiotherapy and conservative non-invasive interventions in scar management?*



*Is there a relationship between the results obtained from a proposed intervention and scar type, scar localization and patient’s age?*



*What are the diagnostic procedures for scar assessment?*



*What is the current evidence regarding the safety of physical therapy modalities in scar management, in terms of adverse reactions, delayed scar maturation or deterioration of scar parameters, and patient tolerance?*


### 2.1. Eligibility Criteria

Only studies in English or Italian were considered. Primary sources were excluded if already incorporated into an included evidence synthesis unless the data they contained were not otherwise reported in the evidence synthesis. No temporal restrictions nor study designs were applied. The Population-Concept-Context” (PCC) strategy is reported in Table 1.

### 2.2. Search Strategy

The search strategy was developed by four reviewers. To ensure rigorous and clinically relevant review outcomes, the research team comprised authors with expertise in evidence synthesis, research methodologies, as well as physiotherapists specialized in scar management (from burns or post-surgery).

A preliminary search strategy was conducted on PubMed to identify relevant keywords and terms. MEDLINE Central, PEDro, Embase, Cochrane Library and Central Register of Controlled Trials (CENTRAL), and CINAHL were searched. In addition, the reference list of included studies and the first 12 pages of Google Scholar were searched. The PubMed search strategy is reported in Table 2 and Table 3. Appendix A reports the search strategies adopted. The PRISMA-S was used to report the search strategies [15]. The search was conducted on 20 May 2024.

### 2.3. Study Selection and Data Extraction

The web-based software platform Rayyan (Rayyan Systems Incorporation, Cambridge, MA, USA) was used for duplicate record removal and the selection process [16]. Three reviewers independently screened all titles and abstracts to select eligible articles, after which full-text screening was performed. In particular, for both selection phases, one reviewer screened all articles, while the remaining two reviewers screened a random percentage of articles. Conflicts were solved by a fourth author. Articles lacking an abstract were automatically moved to the full-text review phase. If full-text articles could not be retrieved, the authors were contacted with a maximum of two attempts on a weekly basis. Appendix A includes a report on all excluded articles with an explanation for the exclusion.

Data were extracted by three reviewers independently, as for the “study selection”, using an ad hoc Microsoft Excel form (Microsoft Corporation, Redmond, WA, USA) developed a priori (Appendix A). The information gathered included the following: (1) study design, authors and year of publication; (2) objective of the study; (3) patient population characteristics, such as sample size, demographic details, scar types and localization; (4) concept related to intervention parameters, such as type of intervention, duration, and frequency, also including feasibility, safety and accessibility; (5) context such as location of care and geographical location of care; and (6) outcomes, relevant results and considerations. 

The extracted data were divided by kind of intervention in relation to the study question to examine the effects of physical therapy on scar treatment. In the case of missing data, authors were contacted with a maximum of two email attempts on a weekly basis. If no response was received, the variable was identified as “Not Reported” if any information was lacking, and as “Unclear” if any data was conflicting or incomplete. If there were several publications released from the same study, only the study with the largest sample size was considered. Regular meetings were conducted to identify any problems and discuss updates to the review process with pre-established time frames. Data were presented using descriptive statistics. First, the types of interventions were discussed, along with the rationale and objective for their use and outcomes recorded. Then, a summary of the effects of the interventions in relation to the scar’s characteristics, localization and subject’s age was created. The findings from this scoping review were condensed and visually represented in an infographic. The images reported in the infographic were created using ChatGPT-4o (OpenAI, San Francisco, CA, USA).

## 3. Results

The initial search retrieved 13,419 results. After duplicate removal and the selection process, 92 studies were included. One more study was identified via reference screening [17]. The study by Wang [18] was included despite considering a patient with infection, this choice was justified by the fact that the patient was allocated to the steroid injections group that were not considered in this study. Figure 1 illustrates the selection process [19].

Most of the included studies were narrative reviews (n = 41, 44%), followed by RCTs (n= 18, (19.3%) and clinical trials (n = 13, 13.9%).

Burn scars and HS were the most considered (n = 79, 84.9%), accounting for 84.9% of the studies. Seven studies did not report scar type (7.5%).

Fifty-three studies considered various body areas (56.9%); the most common regions investigated were the upper limb (n = 27, 29%), lower limbs (n = 20, 21.5%), and hands (n = 16, 17,2%). A minority of studies also considered trunk (n = 13, 13.9%) and face scars (n = 5, 5.3%).

Regarding the age of the population, 27 considered adults (29%), six considered only a pediatric population (6.5%), 25 studies considered both adults and a pediatric patient (26.9%), and 35 did not report age range (37.6%). Table 4 reports the characteristics of the included studies.

### 3.1. Assessment Methods

Among the included research, scar assessment procedures were reported by 48 (51.6%) studies [18,20,21,22,23,24,25,26,27,28,29,30,31,32,33,34,35,36,37,38,39,40,41,42,43,44,45,46,47,48,49,50,51,52,53,54,55,56,57,58,59,60,61,62,63,64,65,66]. Table 5 illustrates the methods used for scar assessment in the included studies.

### 3.2. Interventions

Pressure therapy was the most reported intervention (n = 41, 44.1%) [17,20,21,22,23,24,25,26,27,28,67,68,69,70,71,72,73,74,75,76,77,78,79,80,81,82,83,84,85,86,87,88,89,90,91,92,93,94,95,96,97]. Pressure values ranged from five to 50 mmHg, and the wearing time ranged between 30 min and 24 h a day; treatment duration lasted from three to 18 months. Nine studies considered the patient’s adherence and treatment complications [17,71,73,83,85,86,87,91,96].

Physical therapy modalities were the second most investigated intervention and have been found in 37 studies (39.8%) [18,21,22,23,26,29,30,31,32,33,34,35,36,37,38,39,40,41,42,43,44,45,46,72,74,77,82,84,85,87,88,89,95,98,99,100]. In total, 25 studies considered laser therapy (26.9%) [21,23,26,29,31,32,36,37,38,41,42,43,46,72,77,79,82,84,85,87,88,89,95,99,100], of which 16 analyzed (17.2%) pulsed-dye-laser (PDL) [23,31,37,41,42,46,77,82,84,85,87,88,89,95,99,100], five (5.3%) analyzed intense-pulsed light (IPL) [42,82,84,89,99], four analyzed low-level laser therapy (LLLT) (4.3%) [23,36,38,99], two analyzed high-intensity laser therapy (HILT) (2.1%) [23,43], seven considered Yag laser (7.5%) [23,29,77,79,82,87,89], and five considered non-ablative fractional laser (NAFL) (5.3%) [21,32,41,42,99].

Across the included laser therapy studies, the reported parameters varied widely. For PDL, 1–6 sessions were typically delivered at 4–8 weeks intervals, with pulse durations of 0.45–10 ms, spot sizes of 7–10 mm, and fluences of 3–12 J/cm^2^. IPL protocols generally applied ~40 J/cm^2^, NAFL used 1540 nm/15 ms/70 mJ/cm^2^, LLLT 632.8 nm/16 J/cm^2^, and HILT 1064 nm with 510–1780 mJ/cm^2^. Most protocols were empirically chosen.

Extracorporeal shockwave therapy was reported in six studies (6.4%) [18,33,40,44,84,98], cryotherapy was reported in five studies (5.3%) [23,34,77,85,87], ultrasound therapy was reported in four studies (4.3%) [35,39,74,99], and iontophoresis [30] and light therapy [35] were reported in one study, respectively (1%). One study considered the association between low-intensity electromagnetic and electrical stimulation and negative pressure (1%) [45]. Two studies considered adverse effects [87,95]. The application parameters reported in the studies are illustrated in Table 6.

In total, 29 studies investigated the effects of silicone-based products on scar parameters (31.2%) [20,21,25,26,47,48,49,50,51,52,53,54,55,56,67,71,72,77,79,81,82,84,85,86,87,88,89,95,97]. Silicone Gel Sheeting (SGS) was worn between 12 and 24 h a day, the treatment duration ranged from 45 days to 12 months, and Silicone gel was applied two to four times a day for one to six months [48,49,55,89]. Seven studies considered adverse effects [47,48,77,79,82,87,97].

In total, 20 studies addressed massage interventions (21.5%) that included massage, deep friction massage and soft-tissue mobilizations [22,47,57,58,59,60,61,62,67,70,71,72,74,79,81,84,87,92,101,102]. The time of administration per session ranged from three to 30 min, ranging from one to five daily sessions repeated one to three times a week. Ten studies reported the techniques adopted [22,47,57,58,59,60,71,81,87,92]. The study by Atiyeh was the only one to report adverse effects [87]. Table 7 reports the techniques adopted.

A minority of studies focused on splinting (n = 17, 18.3%) [22,26,63,68,70,71,72,74,75,76,78,81,92,93,94,103,104] and patient education (n = 11, 11.8%) [29,47,67,68,69,70,71,72,101,102,105]. Ten studies (10.7%) considered range of motion (ROM) and stretching exercises [22,63,68,69,71,74,75,76,81,101]. Contraindications were reported in two studies [71,74].

Adhesive tapes were reported in seven studies (7.5%) [21,57,64,65,77,87,106]. The application parameters ranged from 25% to 100% of tape tension; tape was worn 24 h a day for four to nine days in a period between six and 12 weeks. Two studies considered adverse effects and contraindications [87,106].

A study on adults with burns to the lower extremities considered the effects of robot-assisted gait training (RAGT) (1%) [66]. The intervention consisted of stepping in place with forward and backward gait for 30 min per day, five days per week for 12 weeks.

Figure 2 reports the number of studies addressing each intervention.

### 3.3. Summary of Reported Effects

Although this review was not designed to perform a quantitative synthesis of effectiveness, most included studies reported outcomes related to scar characteristics and patient-reported symptoms. Across interventions, the commonly reported benefits included improvements in scar pliability, thickness, and color, as well as reductions in pain and pruritus. For example, pressure therapy was often associated with improved scar maturation and reduced thickness, while silicone-based products were most frequently linked to enhanced pliability and color. Laser therapy studies reported reductions in erythema and scar height, particularly with PDL. However, the heterogeneity of measures and study designs, coupled with small sample sizes, limits the strength of these findings.

## 4. Discussion

The included studies revealed a wide range physical interventions available for scar management. These strategies have been implemented for different types of scars with various characteristics. Most studies considered an adult post-burn population. While a substantial number of studies also analyzed other scar types, very little research has been conducted on a non-adult population. These findings highlight the need for future research to provide a better understanding of this population. Furthermore, more than 50% of the studies focused on scars located in various body areas and of various sizes; future studies should aim to report scar localization in greater detail.

It is also important to underline that the parameters of application varied considerably from a study to another. The standardization of application parameters, also based on scar and patient’s characteristics, is needed to allow consistency between future research among trials. Lastly, adverse effects and contraindications were well documented by the included studies for each intervention.

Considering the included research types, there are some limitations in the literature that can influence the interpretation of results. First, most studies had reduced sample sizes and no control groups. These factors contributed to lowering the external validity of the results obtained both in favor or in contrast to the interventions proposed. Moreover, since almost half of the included studies were based on clinical experience or expert opinions regarding the type and parameters of interventions, there is a strong need for future research to critically evaluate the effects of physical therapy techniques.

### 4.1. Assessment Procedures

Scar assessment is divided into objective and subjective measurements. Objective measurements involve procedures by which the scar is critically evaluated based on quantitative data, while subjective measurements are based on qualitative data.

Emphasis has been placed on investigating specific scar parameters. Scar thickness, height, size, pigmentation, vascularity, pliability, itchiness, and pain were the most frequently reported.

Scales are part of the subjective measurements. Over the years, many clinical scales have been developed for scar assessment. However, Scott [61] reported that clinicians rarely implement standardized outcome measures in their practice.

For the assessment of scar characteristics, the Vancouver Scar Scale (VSS) and its derivatives and the Patient and Observer Scar Assessment Scale (POSAS) were the most reported. VSS has some limitations; it does not consider associated symptoms such as pain and itching or psychosocial factors [42,107]. It was developed for burns, but its use is also justified for linear scars [107].

Regarding POSAS, patients consider minor differences (less than 0.75 on the 1–10 scale) as clinically important for changes in scar quality [108]. Moreover, it is more sensitive than VSS in assessing scar vascularity [109]. Therefore, it represents a useful instrument for scar assessment [107].

Another reported scale was the DLQI. Despite being used in other dermatologic conditions, it needs to be validated in patients with different scar types [107].

The Visuo-Analogic Scale (VAS) was largely used in the assessment of pain and itching in eleven studies [20,21,28,36,39,40,42,43,45,49,55] and was recommended in one systematic review [110].

For itching, the use of the Itch Severity Scale (ISS) shows good test–retest reliability but is quite time consuming [111]. However, none of the included studies considered this tool, while the Itch Man Scale was preferred in one trial [59].

Objective measurements were performed with different equipment; these tools offer a more reliable and accurate analysis compared to subjective scar scales and therefore are particularly useful for research and clinical purposes [112,113]. The ultrasound assessment of scar thickness was the most reported among the included studies [20,24,26,27,28,33,36,38,42,46,54,57,58]. However, some authors suggest that subjective scar scales should still be used in research as they provide a more global assessment [42,113] and allow the measurement of certain variables that cannot be evaluated with objective devices, such as pain and itch. Major barriers include the high cost of objective measurement tools and the technical skills required by the examiner [42,112].

### 4.2. Pressure Therapy

Pressure therapy is one of the most common interventions for scar management and consists of the application of pressure via compression garments. The mechanism is still not fully understood but it is thought to decrease scar blood flow, reduce edema formation, limit protein deposition, including collagen, and promote lysis [67,77,81,85]. The included studies applied pressure both in a hospital setting, especially burn units, and outpatient rehabilitation settings. Despite its wide use in clinical practice, pressure parameters are not supported by strong clinical evidence rather than empirical findings [77]. Pressure application is usually started after epithelization or for wounds that take more than 14 to 21 days to heal [71,79,80,81,92,96].

The treatment duration ranged between three and 18 months. However, multiple authors recommended continuing up to 12 months or until scar maturation [77,79,86,90], even if poor results were observed after six months from epithelization [85,96].

Pressure magnitudes varied widely across the studies. Candy [24] reported that higher levels of pressure (i.e., 20–25 mmHG) led to better results. Most studies suggested adopting pressure levels above 24 mmHg, which corresponds to the capillary pressure. Notably, this value is only supported theoretically, based on an arterial capillary closing pressure of 25 mmHg [17,27,53,77,81,96].

The wearing time for pressure is also heterogeneous. Many studies considered 23 h per day as a reference. However, there are no included studies comparing the effects of different wearing times. Importantly, receiving clear and thorough information on goals and the proper use of pressure garments may be relevant [73]. A shorter wearing time and accurate instructions on goals and the use of garments could lead to better patient adherence and possibly more favorable outcomes.

An important limitation in the application of pressure therapy is the reduced clinical use and poor reliability of pressure measuring equipment, which is also reflected in this scoping review; only four studies measured pressure during treatment [27,80,81,96].

No difference was reported regarding the application of pressure based on scar type, localization, and age of the population, except for face masks for face scars.

### 4.3. Physical Modalities

#### 4.3.1. Laser Therapy

There are many types of laser therapy, such as PDL, IPL, NAFL, HILT and LLLT. The mechanism of action lies in the coagulation of tissue [99].

PDL was the most used device; it was used for HS, hypertrophic burn-related scars, and keloids. Trials found benefits in reducing scar size, pliability, thickness, particularly for erythema [21,23,31,46,77,85,87,89,99], and preventing scar formation [21,46,77,87,95]. A systematic review considering burn scars found that 585 nm PDL had low efficacy, while 595 nm had moderate efficacy [114]. On the other hand, considering HS and keloids, a meta-analysis reported that the certainty of evidence was low to very low due to the small sample sizes, the limited number of studies, unblinded assessments, and attrition bias, all of which reduced the generalizability of the results [41].

The treatment protocols proposed for PDL varied considerably. A consensus reported that most therapists use a fluence of 5–6 J/cm2, pulse width of 1.5 ms, and a 10 mm spot [42]. These values were only based on clinical experience. Interestingly, Manuskiatti found that the effects of PDL seem to be independent of the energy density adopted [37]. However, the small sample size of this study does not allow for firm conclusions. Thus, additional studies are required to define the most appropriate parameters for PDL application [42].

Fewer studies have considered Nd:YAG laser IPL, HLLT and LLLT. Despite the positive results observed, more trials investigating their effects on scar characteristics are needed because of the small samples and the heterogeneous populations considered.

We did not identify studies comparing the application of one device to another; future trials should also investigate head-to-head comparisons to establish which laser modality is more suitable depending on scar type, localization, scar age and skin type.

It is also important to consider the contextual factors affecting the implementation of laser therapy. The scope of practice for physical therapists in applying laser therapy varies considerably across countries and regions, depending on professional licensure regulations and national healthcare policies. In some jurisdictions, physical therapists are allowed to apply laser therapy autonomously, while in others it is restricted to physicians or requires specific prescription. Moreover, accessibility to laser devices can be limited by their high costs, which may limit adoption in clinical practice, especially in resource-limited settings.

#### 4.3.2. Extracorporeal Shockwave Therapy

ESWT was reported to be effective in reducing pain, itching, thickness, and VSS scores, increasing perfusion, and accelerating re-epithelization in post-burn scars [33,44,98]. The mechanism of action is unknown [18]. A recent meta-analysis of RCTs considering post-burn patients suggested a protocol for ESWT application that needs to be further investigated [44].

The six studies considering ESWT involved very different populations regarding scar type, localization, age, cause of scars, and intervention parameters. Most of the research has been conducted on post-burn adult patients, while keloids were only reported by Wang et al. Future research should investigate the effects of ESWT on scars of different etiologies [18].

#### 4.3.3. Cryotherapy

Few studies, mostly narrative reviews, considered cryotherapy. The effect acts by inducing local ischemia with the subsequent necrosis of scar tissue [115]. Cryotherapy was more commonly applied in the treatment of HS and keloids.

Two studies considered scar size, reporting that this intervention is suitable only for small scars [77,87]; only one reported a treatment effect depending on scar age [85]. However, due to the lack of existing evidence for scar size and age, future studies are needed to validate this hypothesis.

#### 4.3.4. Other Modalities

Ultrasound therapy, iontophoresis, polarized light therapy, and the association between electrical stimulation and negative pressure were reported in a small number of studies [30,35,39,45,74,99]. Moreover, all these trials considered a small sample size that could have negatively influenced the results. The paucity of research considering these modalities highlights the need for future research in these areas to establish their effects.

### 4.4. Silicone-Based Products

Silicone is considered a first-line treatment in scar management [47,77,81,87,97] and is recommended for improving scar thickness, color, and pliability [81]. The accepted mechanism of action is thought to rely on hydration and the occlusion of the stratum corneum [77,81,95]; however, there is still debate on this topic [77,95].

The two main products are SGS and silicone gels. As with pressure therapy, the delivery method of silicone-based products is variable. In the included studies, SGS was usually applied between 12 and 24 h a day and started soon after epithelization [21,79,97] for a minimum of two months [77,85,86], while silicone gels were generally applied twice a day [48,49,81,97]. However, the rationale for the different application parameters was not reported, and two included studies also found positive results with shorter application periods [54,55].

Some trials [20,54,56] found SGS to be effective in improving scar parameters, especially thickness in HS and keloids [52,55]. However, two systematic reviews [50,51] highlighted the poor quality of research investigating SGS, indicating the need for future studies in this area.

The included studies did not differentiate the intervention parameters based on scar localization or the age of the population. The only variation noted was the use of silicone gel for areas with high mobility [21,86] or where SGS attachment would be difficult [89] such as the inframammary fold, face, joint regions, or neck [88]. However, this difference has not been investigated in the clinical trials retrieved by our search strategy. Furthermore, one trial did not find differences in the application of silicone gels versus SGS [48]. Future research should consider comparative trials to better establish the effects of these two silicone-based products.

One study reported that adverse effects are more common with the use of SGS than with silicone gels [81].

Many aspects of silicone-based products remain under-researched. Future trials should clarify their effects on scar characteristics, determine optimal application parameters, and compare the benefits and potential adverse effects of SGS versus silicone gels.

### 4.5. Massage

Sixteen out of 20 studies investigating massage were conducted on post-burn patients, with most of them carried out in hospital settings; only three studies included keloids.

Massage is thought to have positive effects on scar thickness, pliability, adhesion, pain, pruritus [57,87,116] and ROM [62,71] in HS and post-burn scars. However, its efficacy is controversial [58,59,72,87,92]. It is believed to work by breaking collagen fibers, thereby aiding in lysis [72,79].

Evidence suggests initiating massage once the healed tissue has gained sufficient tolerability to withstand surface friction [47,70] and that the technique used should vary based on scar localization [87] and the tissue’s strength [92] and degree of maturation [81]. Although various techniques can be applied, none have been validated [87] and the rationale for choosing one technique over another is primarily based on expert or clinician opinion.

Poddighe et al. [60] was the only study to consider soft-tissue mobilizations, reporting positive results on HS characteristics; however, the study lacked a control group and consisted of only 19 subjects.

Many studies considered the patient’s active engagement by instructing them on how to self-massage the scar [22,47,59,62,92]. A typical home program consisted of scar massage performed three to five times per day for approximately three to ten minutes per area [61,70]. However, this dosage has been developed on an empirical basis [61], thus requiring future validation via clinical trials.

Given these considerations, evidence is lacking regarding the best methods for the delivery of massage; the massage techniques and their application parameters have been mainly based on an empirical basis. Furthermore, it was not possible to retrieve data regarding variations in the intervention based on scar type. Our results are in accordance with the findings of Scott [117] who highlighted the limitations of research in this area.

Future RCTs and comparative trials should consider these variables to establish evidence-based application protocols for massage.

### 4.6. Splinting

Positioning and splinting aim to prevent scar contracture [70,92,93,103] after a burn by influencing the realignment of collagen [103]. Anti-contracture positioning and splinting have been reported as being used since the early phases of the rehabilitation process [26,68,71,103]. Accordingly, all of the studies considered post-burn patients, with all but two being conducted in burn centers.

From the included studies, the absence of a standardized wearing schedule was noted for both for adult and pediatric populations [72,93,103]. Some authors suggested that the length of time a splint is worn each day is determined by multiple factors such as scar location, depth, the malleability or density of the scar, the type of joint function, the patient’s physical and psychological tolerance, and the ability to actively move the affected area [68,70,74,81]. Moreover, some studies suggest using different splint types in different phases [70,75,81,93,103]. However, these variables are only expressed through clinical experience and there are no trials identified by our search strategy that investigate these areas.

Regarding scar localization, the most common regions requiring splinting were the hands, elbow, neck, axilla, wrist, and ankle. This is due to the increased likelihood of developing contractures and heterotopic ossification in joints such as the elbow [70,94,103,104].

Although, patients believe that splinting interventions contribute to the recovery process [78], the number of controlled trials, RCTs, and comparative trials for splinting is rather low, and different techniques or types of applications and products are used [81,93,103]. Future trials should establish criteria for clinical use and parameters for application.

### 4.7. Education

All but one study focused on post-burn patients, with most of them conducted in burn units. The delivery of education is largely based on qualitative studies which support the idea that patients and their families should be informed of the patient’s injuries, course of treatment, and self-management to allow better cooperation and compliance, potentially leading to better results [29,47,63,67,68,70,71,72,101].

One systematic review [102] observed that education within a multimodal rehabilitation approach increased the level of burn knowledge and improved hand function in burn patients. However, the effects on scar characteristics were not reported.

To date, the effects and best modalities for delivering patient education are currently unknown, which highlight the need for future high-quality trials.

### 4.8. Stretching and Mobilizations

Mobilizations are used to lengthen scar tissue, helping to prevent joint and ligament stiffening and thus reducing scar contracture [22,71,75,81,101]. The included studies considered stretching and mobilizations only for post-burn patients.

ROM exercises, both passive and resisted, stretching, and proprioceptive neuromuscular facilitation were the interventions identified in the literature. Some studies also reported the combined use of stretching with paraffin to soften and lubricate the skin [68,69,74].

The parameters of applications are not clearly reported and are mainly based on skin blanching and patient’s response [68,69,76,101]. Furthermore, Asadullah et al. [63] were the only authors to consider the scar age in their study, providing early ROM exercises. Two studies reported stretching in children, highlighting critical points in this population, including decreased attention span, limited cognitive reasoning, small body size, activity levels leading to low compliance [69,71].

Most of the studies considering stretching and ROM exercises were represented by qualitative research and were also based on the author’s experience and opinions. High-quality clinical trials are needed to provide robust evidence regarding these interventions.

### 4.9. Adhesive Tapes

Adhesive tape aims to control the wound tension forces and prevent hypertrophic scarring [77,87]. The application of adhesive tape has been reported to contribute to positive effects on scar height, thickness, pliability, color, and mobility.

The included studies considered HS, burn scars and keloids. Post-burn patients were considered in all but two studies which investigated scars of different etiologies [21,65].

Different parameters of application were used. In general, tape was kept for seven days with a tension varying from 25% to 100%.

No specific differences in applications and effects have been found based on scar localization, size and age. Most studies considered different body areas, sizes, and ages, with only a minority reporting the exact modality of application.

Future studies should investigate their effects and specific techniques of application based on scar size, localization, and age. Moreover, it is crucial to further investigate the effects of tape in a non-burn population.

### 4.10. Robot-Assisted Gait Training

One study investigated the effect of robot-assisted gait training on burn patients. Significant results were only found for scar hysteresis, while other scar characteristics remained unchanged [66].

### 4.11. Limitations

To our knowledge, this is the first study to summarize and map the literature on scar management, providing an overview of the available evidence as a precursor for future research.

The inclusion of studies written in English or Italian may have excluded research relevant to the objectives of this scoping review. Moreover, focusing solely on non-invasive interventions may have limited the research group’s ability to consider additional interventions relevant to the clinical practice of physical therapists, such as studies considering dry needling and acupuncture, which fit in this category.

Although the methodological quality of the included studies was not assessed, an exhaustive overview of the currently available interventions was nonetheless provided.

### 4.12. Implications for Practice

The results of this review not only map the available evidence but also allow for a preliminary critical appraisal of current non-invasive interventions for scar management. The heterogeneity of study designs, limited sample sizes, and lack of standardized protocols mean that many approaches remain supported mainly by clinical experience rather than robust trials. These limitations should be considered when interpreting the suggested parameters reported below.

Considering the results obtained, it is possible to suggest some standardized parameters regarding the application of physical interventions. Nevertheless, these recommendations are still to be investigated by high-quality trials. It is essential to highlight that our findings should not be considered formal clinical recommendations, as the study design only allows us to map the existing literature. The application values are reported in Table 8. Apart from the parameters reported in Table 8, the clinical management of scar tissue should also consider specific features such as scar type, localization, malleability, age, skin color, and the patient’s age and compliance. Frequent assessment and monitoring are useful strategies to adjust the intervention according to the patient’s response. Given these considerations, an infographic (Figure 3) was created to assist patients and clinicians with the physical management of scars.

## 5. Conclusions

A variety of non-invasive physical therapy interventions have been reported for the management of scar tissue. Nevertheless, high-quality evidence to support the widespread use of these interventions is lacking, with some being based only on clinical experiences or expert opinions.

Regarding scar assessment procedures, a combination of objective and subjective measurement tools is recommended to enhance the reliability of measurements and to facilitate their implementation in clinical practice.

From a research standpoint, the main limitations found in the included studies were the reduced sample size and the absence of a control group.

Given these considerations, future high-quality controlled trials should aim to better establish the effects of physical therapy interventions on scar parameters and determine the best application parameters according to scar characteristics such as age, localization, and type.

## Figures and Tables

**Figure 1 jcm-14-05920-f001:**
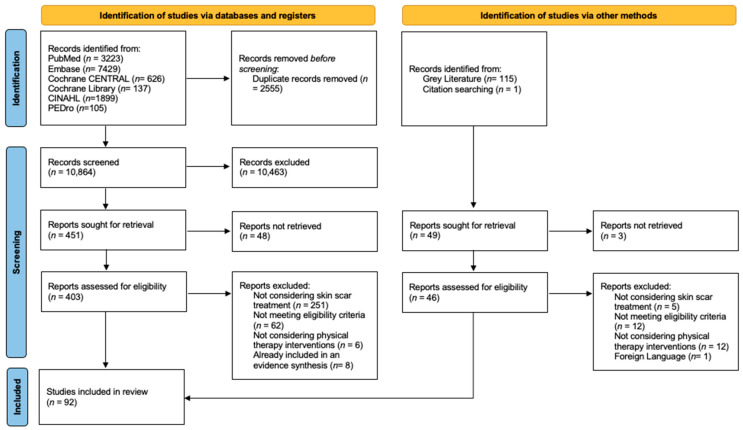
PRISMA 2020 flow diagram.

**Figure 2 jcm-14-05920-f002:**
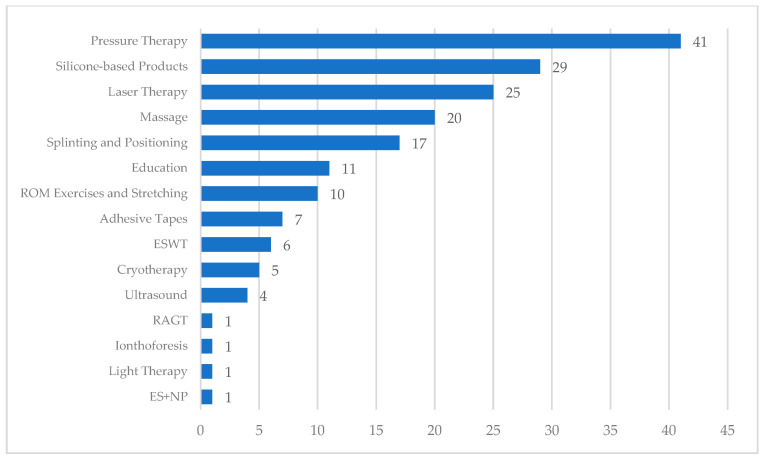
Number of studies per intervention. **Abbreviations: ROM**: Range of Motion; **ESWT**: Extracorporeal Shockwave Therapy; **RAGT**: Robot-Assisted Gait Training; **ES**: Electrical Stimulation, **NP**: Negative Pressure.

**Figure 3 jcm-14-05920-f003:**
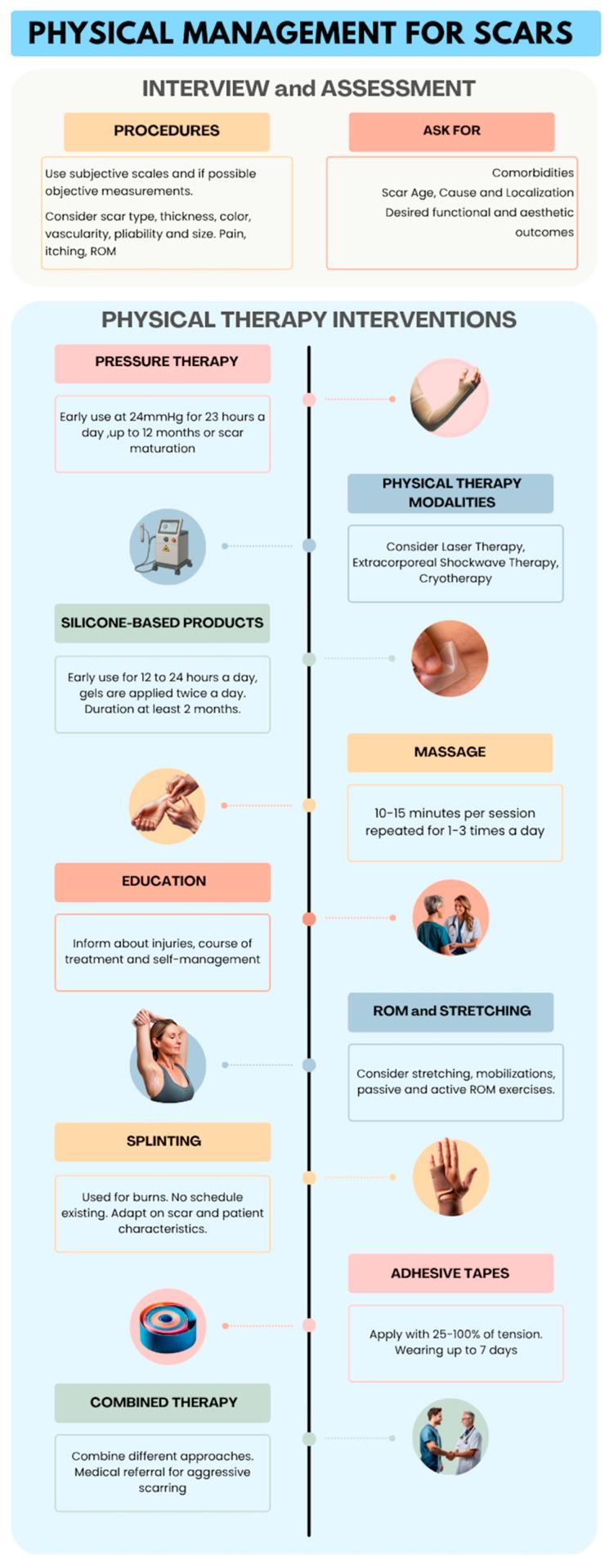
Infographic on the management of scars (AI-generated figure).

**Table 1 jcm-14-05920-t001:** PCC strategy.

Population	All types of scars on humans with no age restrictions were included except for those that interfere with wound healing.
Concept	All types of conservative non-invasive interventions including education and self-management strategies (e.g., scar massage, soft tissue mobilization, splinting, etc.).
Context	Studies were included regardless of geographical location, social or cultural context, or level of care.

**Exclusion Criteria**: animals or in vitro studies, post-operative or trauma infections, use of steroids or chemotherapy, diabetes, restrictive skin disorders or active dermatologic conditions, pregnancy, prior scar surgery, acne scars.

**Table 2 jcm-14-05920-t002:** PCC search terms.

Population (P)	“Cicatrix” [Mesh], “Cicatrix, Hypertrophic” [Mesh], “Tissue Adhesions” [Mesh], “Keloid” [Mesh], “Scar *”, “Scarring”, “Hypertrophic Scar *”, “Burn Scar”, “Contracture Scar”.
Concept (C)	“Rehabilitation” [Mesh], “Exercise Therapy” [Mesh], “Physical Therapy Modalities” [Mesh], “Musculoskeletal Manipulations” [Mesh], “Therapy, Soft Tissue” [Mesh], “Exercise” [Mesh], “Self Care” [Mesh], “Patient education as topic [Mesh], “Muscle Stretching Exercises” [Mesh], “Conservative Treatment [Mesh], “Exercise”, “Habilitation”, “Rehabilitation”, “Physiotherapy”, “Physical Therapy”, “Manual Therapy”, “Patient Education”, “Mobilization”, “Resistance Training”, “Strength Training”, “Stretching”, “Conservative Treatment”, “Non-invasive Treatment”, “Soft Tissue Therapy”.
Context (C)	/

**Table 3 jcm-14-05920-t003:** PubMed Search Strategy.

Database	Query Strings, Keywords and Boolean Operators	Results
Medline (via PubMed)	(((((((((“Scar”) OR (“Scarring”)) OR (“Hypertrophic Scars”)) OR (“Burn Scar”)) OR (“Contracture Scar”)) OR (“Cicatrix” [MeSH Terms])) OR (“Cicatrix, Hypertrophic” [MeSH Terms])) OR (“Keloid” [MeSH Terms])) OR (“Tissue Adhesions” [MeSH Terms])) AND (((((((((((((“Soft Tissue Therapy”) OR (“Rehabilitation”)) OR (“Habilitation”)) OR (“Physiotherapy”)) OR (“Physical Therapy”)) OR (“Exercise”)) OR (“Stretching”)) OR (“Resistance Training”)) OR (“Strength Training”)) OR (“Manual Therapy”)) OR (“Mobilization”)) OR (“Patient Education”)) OR (“Conservative Treatment”))	3223

**Table 4 jcm-14-05920-t004:** Main study features summary.

Variable	Number of Studies (%)
Year of Publication	
1970–1979	1 (1.1%)
1980–1989	7 (7.5%)
1990–1999	5 (54%)
2000–2009	19 (20.4%)
2010–2019	31 (33.3%)
2020–2024	29 (30.1%)
Study Design	
Book Chapter	2 (2.1%)
Clinical Trials	13 (13.9%)
Commentary	1 (1%)
Consensus	3 (3.2%)
Narrative Review	41 (44%)
Observational Study	3 (3.2%)
Prospective Study	2 (2.1%)
RCT	18 (19.3%)
Retrospective Study	1 (1%)
Survey	3 (3.2%)
Systematic Review	5, 2 meta-analyses (5.3%)
Scar Type	
Burn Scars	47 (50.5%)
Contracture	7 (7.5%)
Hypertrophic Scars	32 (34.4%)
Keloid	20 (21.5%)
Not Reported	7 (7.5%)
Population	
Adults	27 (29%)
Children	6 (6.5%)
Adults and Children	25 (26.9%)
Not Reported	34 (37.6%)
**Scar Localization**	
Face	5 (5.3%)
Hand	16 (17.2%)
Lower Limb	20 (21.5%)
Neck	8 (8.6%)
Trunk	13 (13.9%)
Upper Limb	27 (29%)
Various Body Areas	53 (56.9%)
Interventions	
Pressure Therapy	41 (44.1%)
Physical Therapy Modalities	37 (39.8%)
Silicone-based Products	29 (31.2%)
Massage	20 (21.5%)
Splinting	17 (18.3%)
Therapeutic Education	11 (11.8%)
Range of Motion and Stretching Exercises	10 (10.7%)
Adhesive Tapes	7 (7.5%)
RAGT	1 (1%)

**Table 5 jcm-14-05920-t005:** Scar assessment methods from the included studies.

Subjective Tools	Number of Studies	Objective Tools	Number of Studies
Vancouver Scar Scale	23 (24.7%)	Ultrasonography	13 (13.9%)
Visuo Analogic Scale	11 (11.8%)	Doppler Laser or Ultrasound	4 (4.3%)
Patient and Observer Scar Assessment Scale	9 (9.7%)	Spectrocolorimeter^®^	3 (3.2%)
Modified Vancouver Scar Scale	5 (5.4%)	Colormeter^®^	2 (2.1%)
Dermatology Life Quality Index	2 (2.1%)	DermaLab Elasticity Probe	2 (2.1%)
Smith Scale	2 (2.1%)	Cutometer^®^	2 (2.1%)
Hamilton Scale	2 (2.1%)	Mexameter^®^	2 (2.1%)
Manchester Scar Scale	2 (2.1%)	Chromameter^®^	2 (2.1%)
Seattle Scale	1 (1%)	Tonometer^®^	1 (1%)
Vancouver Burn Scale	1 (1%)	Pressure Pain Treshold	1 (1%)
Modified Itching Severity Scale	1 (1%)	Tewameter^®^	1 (1%)
Itch Man Scale	1 (1%)	3D Scanner	1 (1%)
Numeric Rating Scale	1 (1%)		

**Table 6 jcm-14-05920-t006:** Physical therapy modality application parameters.

Modalities	Parameters
PDL	585–595 nm, 3 to 12 J/cm^2^, 0.45–10 ms, 7–10 mm spot for 2–6 sessions
NAFL (one study)	1540 nm, 15 ms pulse duration, 70 mJ/cm^2^
LLLT	632.8 nm, 119 mW/cm^2^, energy density was 16 J/cm^2^ for 25 min in 24 sessions.
IPL	515–1200 nm, 40 J/cm^2^
HILT:	1064 nm, 3 kW, 510–1780 mJ/cm^2^, 10–40 Hz, 120–150μs, duty cycle of about 0.1% for 18 sessions over 3 weeks
Ultrasound (one study)	1 MHz, 1 w/cm^2^, 10 min. Every day for 2 weeks
ESWT	100 to 3000 pulses, 0.015 to 0.3 mJ/mm^2^, 1 to 6 Hz, 1 to 12 sessions
Ionthoforesis (one study)	2.5–4 mA, 40–50 mA/min., 3 to 12 sessions
Cryotherapy (one study)	10 min, −14 degrees, 2 sessions per week, for 10 weeks
Light therapy (one study)	480–3.400 nm, degree of polarization >95% (590–1550 nm), 40 mW/cm^2^, light energy per minute 2.4 J/cm^2^. 30 min, three times a week for 4 weeks
ES + NP	5 Hz square wave with adjustable output up to 0.36 mA at 500 Ohm. Negative pressure up to 35 atm.

**Abbreviations: PDL:** pulsed-dye-laser; **NAFL**: nonablative fractional laser; **LLLT**: low-level laser therapy; **IPL**: intense pulsed light; **HILT**: high-intensity laser therapy; **ESWT**: extracorporeal shockwave therapy; **ES**: electrical stimulation; **NP**: negative pressure.

**Table 7 jcm-14-05920-t007:** Techniques used for scar massage.

Techniques
Cutaneous mobilizations
Pulpar massage
Push-pull
J Stroke
Modified version of the “Indian Burn”
Skin Rehabilitation Massage Therapy
Deep friction massage
GAF techniques
Massage dermo-épidermique
Pressing movements
Circular, transverse, and vertical strokes
Pinching and lifting

**Table 8 jcm-14-05920-t008:** Suggested parameters for physical intervention application.

Interventions	Suggested Parameters
Pressure Therapy	Early application at 24 mmHg for 23 h a day up to 12 months or scar maturation.
ESWT for burn scars	1–2 sessions per week for 4–8 weeks, at least 100 ESWs per cm^2^ or between 2000 and 3000 ESWs in total per session, highest EFD the patient can tolerate.
Silicone-based Products	*SGS*: Early application, Worn between 12 and 24 h a day and maintained at least for 2 months *Silicone Gel*: early application, two times per day at least for 2 months
Massage	10–15 min per session repeated for 1–3 times during the day.
Splinting	Application should vary considering localization, depth, elasticity of the scar and patient’s tolerance and ability to move the affected area
Education	Inform about injuries, course of treatment and self-management.
Stretching and Mobilizations	Stretching based on skin blanching and patient’s tolerance. Active, passive, resisted ROM exercise and Proprioceptive Neuromuscular Facilitations.
Adhesive Tapes	25–100% tension, worn for 7 days.

**Abbreviations**: **ESWT**: Extracorporeal Shockwave Therapy; **ESWs**: Extracorporeal Shockwaves; **EFD**: Energy Flux Density; **SGS**: Silicone Gel Sheeting.

## Data Availability

Data are available in the Appendix A of this article. The authors can provide further material upon reader request.

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
