# Peer review of "Current Physical Therapy for Skin Scar Management: A Scoping Review"

_jcm, 2025, doi:10.3390/jcm14175920_

Round 1

Reviewer 1 Report

Comments and Suggestions for Authors

  • Opening statement focuses on high income countries; is there any data available for low income- may speak more to the global disability from scars?

Methods

  • Good- clear, concise

Results

  • Tables and charts create excellent visualizations of included studies •For laser section: all other modalities noted in results section discuss parameters (dose, settings, frequency, etc.); is it possible to discuss parameters for laser section?

Discussion

  • Overall, each section tells a good “story” with flow of method of action, application, and evidence
  • Laser- if relevant, comment on the scope of practice of laser in physical therapy- this likely differs between countries/regions. Also could comment on accessibility of laser therapy and cost •Move assessment procedures to beginning of discussion sections- coincides better with how you treat a patient- assessment then treatment

Overall

  • Overall I think the study is well written and tells a story with good flow, especially through the discussion section of providing an overview of the modality, use and evidence found •May I suggest taking on the mindset of who your “goal reader is”- ie. A practitioner/PT/OT would have a preference for dosage and applications (where your discussion section focuses), vs. researcher (data management. Make this mindset clear throughout the paper.

Comments on the Quality of English Language

english needs to be modified to a target audience and a bit more focused.

Hypotyhesis driven and with a clearer goal.

plse rewrite.

Author Response

Response to Reviewer 1 Comment

Comments and Suggestions for Authors

  • Opening statement focuses on high income countries; is there any data available for low income- may speak more to the global disability from scars?

We appreciate this suggestion. We have now expanded the opening statement of the Introduction to briefly address the burden of scars in low- and middle-income countries. The following addition “In low- and middle-income countries, scars also represent a substantial burden, with limited access to specialized care and rehabilitation services contributing to worse functional outcomes, greater disability, and socioeconomic impact” highlights the role of limited access to specialized care and rehabilitation in exacerbating disability and socioeconomic impact in these settings, aligning the introduction with a more global perspective.

  • Result: For laser section: all other modalities noted in results section discuss parameters (dose, settings, frequency, etc.); is it possible to discuss parameters for laser section?

We thank the reviewer for this valuable observation. In response, we have integrated a concise summary of the reported parameters for laser therapy in the Results section (3.1 Interventions), following the same format used for other modalities. This addition includes typical ranges for PDL, IPL, NAFL, LLLT, and HILT protocols, highlighting the variability across studies and the current lack of standardization. We added “Across the included laser therapy studies, reported parameters varied widely. For PDL, 1–6 sessions were typically delivered at 4–8 week intervals, with pulse durations of 0.45–10 ms, spot sizes of 7–10 mm, and fluences of 3–12 J/cm². IPL protocols generally applied ~40 J/cm², NAFL used 1540 nm/15 ms/70 mJ, LLLT 632.8 nm/16 J/cm², and HILT 1064 nm with 510–1780 mJ/cm². Most protocols were empirically chosen.”

  • Discussion: Overall, each section tells a good “story” with flow of method of action, application, and evidence. Laser- if relevant, comment on the scope of practice of laser in physical therapy- this likely differs between countries/regions. Also, could comment on accessibility of laser therapy and cost

We thank the reviewer for this valuable suggestion. We agree that scope of practice, accessibility, and cost are important contextual factors for the application of laser therapy. We have therefore added a short statement in the Laser Therapy subsection of the Discussion to highlight that the use of laser in physical therapy varies across countries and may be influenced by national regulations, availability of equipment, and economic factors. The following addition “It is also important to consider contextual factors affecting the implementation of laser therapy. The scope of practice for physical therapists in applying laser therapy varies considerably across countries and regions, depending on professional licensure regulations and national healthcare policies. In some jurisdictions, physical therapists are al-lowed to apply laser therapy autonomously, while in others it is restricted to physicians or requires specific prescription. Moreover, accessibility to laser devices can be limited by their high costs, which may limit adoption in clinical practice, especially in re-source-limited settings.” aims to provide a broader perspective for clinicians and policymakers interpreting the findings.

  • Move assessment procedures to beginning of discussion sections- coincides better with how you treat a patient- assessment then treatment

According to the reviewer comment we moved the assessment procedures to beginning of discussion sections.

  • May I suggest taking on the mindset of who your “goal reader is”- ie. A practitioner/PT/OT would have a preference for dosage and applications (where your discussion section focuses), vs. researcher (data management. Make this mindset clear throughout the paper.

We appreciate the reviewer’s insightful suggestion. The primary intended audience of this scoping review includes practitioners (PTs) and clinicians involved in scar management. In response, we have clarified this focus (also according following comments from both reviewers) in the Introduction and ensured that dosage, application parameters, and clinical implications are explicitly reported in the Results and Discussion sections for all interventions. Where applicable, we maintained concise methodological details for research purposes but emphasized practical, clinically relevant information to enhance the paper’s utility for healthcare professionals. we have now added “Given that the primary intended audience of this review includes clinicians such as physiotherapists involved in scar management, we emphasized dosage, application parameters, and clinical implications to support evidence-based decision-making in daily practice, while still providing sufficient methodological detail for research purposes.” In the last paragraph of the introduction.

  • Comments on the Quality of English Language: english needs to be modified to a target audience and a bit more focused.

The entire manuscript was reviewed to enhance English language, more clarity and audience targeting according to the reviewers’ comments.

  • Methods: Good- clear, concise
  • Results: Tables and charts create excellent visualizations of included studies
  • Overall I think the study is well written and tells a story with good flow, especially through the discussion section of providing an overview of the modality, use and evidence found.
  • Hypotyhesis driven and with a clearer goal.

We would like to thank the reviewer for the comments and for the opportunity to respond point-by-point to the feedback raised during the revision of our manuscript. The reviewer’s comments were extremely helpful in improve our article. We are also glad for the positive comments.

Reviewer 2 Report

Comments and Suggestions for Authors

The Article “Current Physical Therapy for Skin Scar Management: A Scoping Review” (Manuscript ID: jcm-3804997) summarizes physical therapy modalities used to treat skin scars through a systematic review. The methodology of this study is relatively reliable, with search keywords covering all major complementary and alternative medicine (CAM) therapies, and the results are presented in considerable detail. However, certain issues require supplementation. I detail my critiques as follows:

  1. This article has detailed the findings of the included studies, such as demographic characteristics and the number of reports for each therapy. But, for both clinicians and policymakers, efficacy should be the foremost concern. I was unable to find a presentation of this information in the results section.
  2. The discussion section's enumeration of the length demonstrates the impressive workload undertaken by the author team. However, rather than a mere recitation, I expected to find the authors' critical assessment of these therapies in the discussion.

Comments on the Quality of English Language

Grammar, formatting, and spelling also require improvement to enhance readability.

Author Response

Response to Reviewer 2 Comment

The Article “Current Physical Therapy for Skin Scar Management: A Scoping Review” (Manuscript ID: jcm-3804997) summarizes physical therapy modalities used to treat skin scars through a systematic review. The methodology of this study is relatively reliable, with search keywords covering all major complementary and alternative medicine (CAM) therapies, and the results are presented in considerable detail. However, certain issues require supplementation. I detail my critiques as follows:

We would like to thank the reviewer for the comments and for the opportunity to respond point-by-point to the issues raised during the revision of our manuscript. The reviewer’s comments were extremely helpful in improve our article. We hope that the reviewer will find our changes satisfactory.

This article has detailed the findings of the included studies, such as demographic characteristics and the number of reports for each therapy. But, for both clinicians and policymakers, efficacy should be the foremost concern. I was unable to find a presentation of this information in the results section.

We thank the reviewer for this important observation. We agree that a concise synthesis of the reported effects of the interventions is essential for clinicians and policymakers. While study design scope and the primary aim of this scoping review was to map the available evidence rather than to quantitatively evaluate effectiveness, we have now added a dedicated paragraph in the Results section entitled “3.3 Summary of reported effects.” Reporting the following “Although this review was not designed to perform a quantitative synthesis of effectiveness, most included studies reported outcomes related to scar characteristics and patient-reported symptoms. Across interventions, commonly reported benefits included improvements in scar pliability, thickness, and color, as well as reductions in pain and pruritus. For example, pressure therapy was often associated with improved scar maturation and reduced thickness, while silicone-based products were most frequently linked to enhanced pliability and color. Laser therapy studies reported reductions in erythema and scar height, particularly with PDL. However, the heterogeneity of measures and study designs, coupled with small sample sizes, limits the strength of these findings.”. This paragraph narratively synthesizes the main efficacy-related outcomes reported across the included studies, highlighting the most frequently observed benefits and limitations for each intervention type. This addition should make the available information on efficacy more visible and accessible to the reader.

The discussion section's enumeration of the length demonstrates the impressive workload undertaken by the author team. However, rather than a mere recitation, I expected to find the authors' critical assessment of these therapies in the discussion.

We thank the reviewer for this observation. While the “Implications for practice” section already included a synthesis of the main take-home messages, we have now enriched it with an explicit introductory statement that critically frames the current evidence base, as follow “The results of this review not only map the available evidence but also allow for a preliminary critical appraisal of current non-invasive interventions for scar management. The heterogeneity of study designs, limited sample sizes, and lack of standardized protocols mean that many approaches remain supported mainly by clinical experience rather than robust trials. These limitations should be considered when interpreting the suggested parameters reported below.”. This addition emphasizes the methodological heterogeneity, limited sample sizes, and reliance on clinical experience for many interventions, clarifying the context in which the suggested parameters should be interpreted. We believe this revision makes the critical appraisal more visible and directly addresses the reviewer’s comment.

Comments on the Quality of English Language: Grammar, formatting, and spelling also require improvement to enhance readability.

The entire manuscript was reviewed to enhance English language, more clarity and audience targeting according to the reviewers’ comments.